# Effects of Spiro-Cyclohexane Substitution of Nitroxyl Biradicals on Dynamic Nuclear Polarization

**DOI:** 10.3390/molecules27103252

**Published:** 2022-05-19

**Authors:** Nargiz B. Asanbaeva, Larisa Yu. Gurskaya, Yuliya F. Polienko, Tatyana V. Rybalova, Maxim S. Kazantsev, Alexey A. Dmitriev, Nina P. Gritsan, Nadia Haro-Mares, Torsten Gutmann, Gerd Buntkowsky, Evgeny V. Tretyakov, Elena G. Bagryanskaya

**Affiliations:** 1N.N. Vorozhtsov Institute of Organic Chemistry, Siberian Branch of Russian Academy of Sciences (SB RAS), 9 Acad. Lavrentiev Avenue, Novosibirsk 630090, Russia; nargiz-asan@mail.ru (N.B.A.); gurlar82@nioch.nsc.ru (L.Y.G.); polienko@nioch.nsc.ru (Y.F.P.); rybalova@nioch.nsc.ru (T.V.R.); kazancev@nioch.nsc.ru (M.S.K.); 2V.V. Voevodsky Institute of Chemical Kinetics and Combustion, SB RAS, 3 Institutskaya Str., Novosibirsk 630090, Russia; dmitralek@rambler.ru (A.A.D.); gritsan@kinetics.nsc.ru (N.P.G.); 3TU Darmstadt, Eduard-Zintl-Institute for Inorganic and Physical Chemistry, Alarich-Weiss-Straße 8, 64287 Darmstadt, Germany; haromares@chemie.tu-darmstadt.de (N.H.-M.); gutmann@chemie.tu-darmstadt.de (T.G.); gerd.buntkowsky@chemie.tu-darmstadt.de (G.B.); 4N.D. Zelinsky Institute of Organic Chemistry, Russian Academy of Sciences, 47 Leninsky Prosp., Moscow 119991, Russia; tretyakov@ioc.ac.ru

**Keywords:** EPR, nitroxide, biradical, ferrocene, exchange interaction, DNP

## Abstract

Spiro-substituted nitroxyl biradicals are widely used as reagents for dynamic nuclear polarization (DNP), which is especially important for biopolymer research. The main criterion for their applicability as polarizing agents is the value of the spin–spin exchange interaction parameter (*J*), which can vary considerably when different couplers are employed that link the radical moieties. This paper describes a study on biradicals, with a ferrocene-1,1′-diyl-substituted 1,3-diazetidine-2,4-diimine coupler, that have never been used before as DNP agents. We observed a substantial difference in the temperature dependence between Electron Paramagnetic Resonance (EPR) spectra of biradicals carrying either methyl or spirocyclohexane substituents and explain the difference using Density Functional Theory (DFT) calculation results. It was shown that the replacement of methyl groups by spirocycles near the N-O group leads to an increase in the contribution of conformers having *J* ≈ 0. The DNP gain observed for the biradicals with methyl substituents is three times higher than that for the spiro-substituted nitroxyl biradicals and is inversely proportional to the contribution of biradicals manifesting the negligible exchange interaction. The effects of nucleophiles and substituents in the nitroxide biradicals on the ring-opening reaction of 1,3-diazetidine and the influence of the ring opening on the exchange interaction were also investigated. It was found that in contrast to the methyl-substituted nitroxide biradical (where we observed the ring-opening reaction upon the addition of amines), the ring opening does not occur in the spiro-substituted biradical owing to a steric barrier created by the bulky cyclohexyl substituents.

## 1. Introduction

Dynamic nuclear polarization (DNP) recently became the main method for increasing the sensitivity of solid-state NMR spectroscopy, and this enhancement is especially important in the study of biological systems. DNP [1,2,3,4,5,6,7] is based on the transfer of polarization from strongly polarized unpaired electron spins of polarizing agents (PAs) to nuclear spins (e.g., ^1^H or ^13^C) of samples under the action of microwave radiation. Nitroxides, trityls [8,9], biradicals [10,11,12,13], and transition metal complexes [14] have found their applications as PAs. The greatest improvement of DNP characteristics has been achieved by nitroxide biradicals, owing to the cross-effect mechanism of polarization transfer [15,16]. In this mode, DNP performance is strongly correlated with electron and nuclear spin relaxation times, and longer relaxation times typically result in better enhancement [17]. Nitroxide biradicals have been extensively studied in high magnetic fields, and their structure has been improved by varying the length and rigidity of the organic linker to optimize the electron–electron interaction and to tune their functional groups for extending their electronic relaxation time [18,19]. It has been found that the replacement of the gem-dimethyl groups adjacent to the nitroxide moiety by spirocyclohexyl groups eliminates the main spin echo dephasing mechanism and makes the spin echo dephasing time sufficiently long, at temperatures up to ~125 K [20]. By modifying the radical centers with bulky spirosubstituents, the DNP performance could be further increased. Biradicals of the TEKPOL and AMUPol series demonstrated the best enhancements, approaching the theoretical limit of DNP enhancement [11,17,21].

To generate biradicals with a new type of rigid linker, we recently employed intramolecular [2+2]-cycloaddition of ferrocene-1,1′-diyl bis(carbodiimide) derivatives formed by the interaction of 1,1′-bis(triphenylphosphoranylidenamino)-ferrocene with isocyanates [22]. Within the framework of this approach, a corresponding pyrroline nitroxide was selected as a spin-bearing group because of its chemical stability and synthetic availability, as well as the existing knowledge about its electronic and molecular structure [23]. Moreover, this synthetic pathway was applied to targeted synthesis of spin-labeled 2,4-diimino-1,3-diazetidine species containing two different spin carriers: namely, nitronyl nitroxide and pyrroline-*N*-oxyl radicals.

In this paper, to further generalize this approach, we used spirocyclohexane-substituted nitroxides instead of tetramethyl-substituted derivatives. It is known that a spirocyclohexane-substituted nitroxide manifests long-term electron spin relaxation and high stability in reducing media [20,24,25]. Here, we compared the structure and properties among biradicals with methyl and spirocyclohexane substituents at positions 2 and 5 of nitroxides, and found that spirocyclohexane substituents affect not only the magnitude of the exchange interaction but also the reactivity of the 1,3-diazetidine linker. In 1989, Molina et al. [26] demonstrated that bis(arylimino)-substituted 1,3-diazetidines can undergo a ring-opening reaction upon the addition of amines, resulting in the formation of pentasubstituted biguanides. Ring opening under the action of a nucleophile can cause a change in the backbone of a molecule, an increase in the distance between nitroxide moieties, and consequently, an alteration of the spin–spin interaction. It is worth noting that recent DNP studies highlighted the critical role of *J* and D couplings (in particular the *J*/D ratio) in achieving significant enhancements at magnetic fields >9.4 T [27]. Therefore, it is important to comprehensively analyze the influence of various structural factors of the biradical on the DNP enhancement.

Herein we report the findings of our investigation into the effect of a spirocyclohexane substitution in a nitroxide biradical on the exchange interaction, polarizing ability, and reactivity in terms of the ring opening. Two ferrocene-containing nitroxide biradicals were studied by EPR spectroscopy, X-ray diffraction (XRD) analysis, DNP-NMR measurements, and DFT calculations.

## 2. Results and Discussion

### 2.1. Synthesis and Crystal-Structure Characterization of Biradicals ***3*** and ***4***

Biradicals **3** and **4** were prepared according to a procedure from the literature [22]: namely, via an aza-Wittig reaction of bis(triphenylphosphoranylidenamino)-ferrocene with the corresponding 3-isocyanato-2,5-dihydro-1*H*-pyrrol-1-oxyl (compound **1** or **2**). The reaction initially led to the formation of the bis(carbodiimide) intermediate, which then underwent intramolecular [2+2]-cycloaddition, thereby giving desired paramagnets **3** and **4** (Figure 1). The target product was characterized by NMR ^1^H, infrared (IR), and mass spectroscopy. Synthetic details are given in the Experimental Section.

Single-crystal XRD analysis revealed that biradical **4** crystallizes from a mixture of chloroform and heptane in the form of a solvate with chloroform (Figure 1). The crystals have a triclinic *P*-1 space group with an asymmetric unit containing one biradical per solvate molecule (CCDC 2168987). Two cyclopentadienyl (Cp) rings of the ferrocenyl moiety are not parallel; the angle between the planes of the Cp rings is 12.4°. The 2,4-diimino-1,3-diazetidine moiety is planar within ±0.037 Å and is almost perpendicular to the Cp rings of ferrocenyl. In nitroxide **4**, the dihedral angle between the planes of the 2,4-diimino-1,3-diazetidine moiety and pyrroline ring (N5C3C4C5C6) is ~12.4°, whereas in the dimethyl derivative [22] (**3**), this angle is 4.8°. Such planarized conformation of biradical **4** is favorable for the formation of a weak intramolecular hydrogen bond C4–H...N1 [H…N 2.61, C…N 3.166(4) Å, C–H…N 119°] (see details in Appendix A). The dihedral angle between the planes of the 2,4-diimino-1,3-diazetidine moiety and the other pyrroline ring is 61.6°, i.e., it is almost the same as that in biradical **3**. As to bond lengths of NO groups, they are typical [28]: 1.270(4) and 1.275(4) Å for N5–O1 and N6–O2, respectively.

The electrochemical properties of the biradical **4** were investigated by cyclic voltammetry (CV) in a dichloromethane solution, as was previously done for the biradical **3** in ref. [21], and these data were added to Figure 2 for comparison. Cyclic voltammogram of biradical **4** (Figure 2) demonstrated two merged reversible oxidation peaks (E_1/2_ about 0.28 V), which were assigned to oxidation of ferrocene and nitroxide moieties and were observed for biracial **3** at E_1/2_ = 0.26 V and E_1/2_ = 0.48 V, respectively (Figure 2) [22]. Such electrochemical behavior of **4** most probably could be assigned to the more planar conformation and weak intramolecular hydrogen bonds lowering its nitroxide Red/Ox potential. 

### 2.2. EPR Analysis

To characterize local structure of the paramagnetic centers and its dynamics in biradicals **3** and **4**, we conducted X-band (9.4 GHz, 0.35 T) continuous-wave (CW) EPR experiments. Figure 3 shows the CW EPR spectra of diluted degassed toluene solutions of biradicals **3** and **4**, as recorded at different temperatures.

Figure 3 indicates that the recorded EPR spectra refer to biradicals dissolved in a nonviscous solvent and containing two identical nitroxide radical parts (characterized by the hyperfine interaction [HFI] constant with one ^14^N atom, A_N_) that are coupled by a moderate exchange interaction [29]. The spectra contained a nitroxide triplet with a central line and two outermost lines of the same intensity and interline splitting equal to A_N_ = 1.42–1.44 mT. In the *EasySpin* software [30], the liquid EPR spectra in Figure 3 were fitted to the exchange Hamiltonian expressed as
(1)$$\hat{H}=-2J{\hat{\vec{S}}}_{1}{\hat{\vec{S}}}_{2}$$
under the assumption that exchange interaction parameters *J* were distributed according to the Gaussian function with standard deviation σ*_J_*.
(2)$$f_{J}=\frac{1}{\sqrt{2\pi }\sigma_{J}}\exp\left(\frac{4{\left(J-J_{0}\right)}^{2}}{2\sigma_{J}^{2}}\right)$$

The experimental EPR spectra of biradical **3** in the temperature range 290–370 K proved to be well reproduced with A_N_ = 1.44 mT and the parameters *J*_0_ and σ*_J_* given in Table 1. Readers can see that with increasing temperature, parameter *J*_0_ monotonically shifts toward higher values.

In contrast to biradical **3**, the spectra of biradical **4** in the temperature range 290–330 K were well reproduced under the assumption of two types of biradicals in almost equal amounts with the same HFI constants a_N_ = 1.42 mT but with different exchange interaction parameters *J*: 0 and 218 MHz (Table 2). At temperatures above 350 K, the EPR spectra of biradical **4** can be well reproduced under the assumption of a Gaussian distribution of parameters *J*, similarly to the case of biradical **3** (Table 2).

Results of the simulation of the EPR spectra (Table 1 and Table 2) can be explained for both radicals by the presence of many conformers in solution with noticeably different values of parameters *J* or even with a wide distribution over *J*. The presence of a single type of conformers in the crystal follows from the XRD data. Nonetheless, several conformers can exist in equilibrium in solution. To determine which biradical conformers coexist in solutions, quantum chemical calculations were carried out next.

### 2.3. Quantum Chemical Calculations

At the first step, geometrical parameters of biradicals **3** and **4** were optimized, starting from the geometries extracted from the XRD data. As expected, the optimized geometry is very close to XRD data. Table 3 contains the results of calculations of g_iso_, zero-field splitting parameters (ZFS, D, and E/D), and HFI constants a_N_ for the optimized geometries of **3** and **4**. Table 3 shows that biradicals **3** and **4** have very similar spin Hamiltonian parameters, and the HFI constants are significantly underestimated (by ~30%); this problem is typical for DFT calculations of HFI constants in nitroxides [31,32].

It is expected that the pyrroline rings in solution can rotate around single bonds N3–C and N5–C (see Figure 1 for the numbering) for both biradicals **3** (Figure 4) and **4** (Figure 5). This property in turn will induce conformational transitions and equilibration of several conformers that differ in the distances between the paramagnetic centers (NO groups) and in their mutual orientation and consequently in the magnitude of the exchange interaction parameters.

Based on the optimized structure of biradical **3** in the triplet state, we performed energy scans for the rotation of the pyrroline rings. For the rotation around the N3–C bond, DFT calculations predicted two minima on the potential-energy surface (Figure 4D) with the same energy and separated by a sufficiently low barrier (4.2 kcal/mol). During the rotation around the N5−C bond, three energy minima were found. Nonetheless, the third minimum has rather high energy (~5.5 kcal/mol) and is not populated in the temperature range under study. The energy difference between the other two conformers is negligible, and the interconversion barrier is low (2.1 kcal/mol). The fourth conformer corresponds to rotation of both pyrroline rings. Therefore, according to the computations, all the conformers that arise during the rotation around each bond alone are in equilibrium and are very similar in energy.

Exchange interaction parameters *J* for these conformers were estimated at the BS-DFT level (Figure 4A,C,E). Note, however, that even numerical accuracy of the calculations (~0.004 cm^−1^ or 120 MHz) is close to the experimental values of *J* (Table 1 and Table 2). Thus, we can make only a qualitative comparison of the theoretical and experimental data. The computation showed that there are only four conformers with significantly different *J* values, which cannot explain the Gaussian distribution of the experimental values. One can only assume that the presence of such a distribution can be attributed to the population of the vibrational levels corresponding to the potentials given in Figure 4B,D. In the case of strong anharmonicity, the average distances between NO groups and their mutual orientations in vibrationally excited states should differ from those in the ground state, thus leading to a change in *J*.

Similar calculations were performed for biradical **4** (Figure 5). As in the case of **3**, four low-energy minima corresponding to conformers were identified. *J* values of two of them (Figure 4C,E) are an order of magnitude higher than those of the other two (A and its counterpart), and the latter are less than the numerical accuracy of the calculations. Very low *J* for two conformers and much higher values for the two others are in agreement with the data in Table 2, pointing to the existence of two sets of biradical **4** with *J* ≈ 0 and ≈ 200 MHz. The appearance of the *J* distribution at T ≥ 330 K can also be explained by the population of the vibrational sublevels. The manifestation of this effect at elevated temperatures may be ascribed to a higher barrier for **4** (Figure 4D), resulting in either higher vibrational frequencies in this potential or lower anharmonicity. Unfortunately, it is not possible to test our hypotheses by calculations.

### 2.4. DNP Experiments

#### 2.4.1. Enhancements in ^1^H Magic Angle Spinning (MAS) and ^1^H→^13^C Cross-Polarization Magic Angle Spinning (CPMAS) Experiments

In Figure 6, ^1^H MAS and ^1^H→^13^C CPMAS data are shown for radicals **3** and **4**. In all the spectra, the signal from the silicon plug at ~0 ppm is marked with #. Biradical **3** manifests an enhancement of 15 for ^1^H and of 14 for ^13^C (Figure 6a,b), meanwhile polarization agent **4** achieved an enhancement of only 4 for both nuclei. The signal enhancements almost identical between ^1^H MAS and ^1^H→^13^C CPMAS for each radical clearly indicate a homogeneous distribution of the radicals in a tetrachloroethane (TCE) solution yielding homogeneous spread of the polarization through the sample. The difference in the DNP enhancements between the two radicals may be explained by the presence of a considerable proportion (~50%) of the conformer of biradical **4** with *J* = 0 MHz, which reduces the DNP effect as compared to radical **3**. As already noted, *J* has a major influence on the DNP effect. In **3** and **4**, *J* is modulated not only by conformation transitions but also by specific chemical properties inherent in 1,3-azetidines. The latter are known to undergo ring cleavage upon a reaction with nucleophiles. This transformation gives rise to new biradical(s) with greater distances between the spin moieties and very low *J*. Therefore, it is also important to examine the reaction of biradicals with nucleophiles in more detail (see Section 2.5).

#### 2.4.2. H Build-Up Curves

DNP build-up curves (the dependence of signal intensity on time of microwave irradiation) for ^1^H at nominal 115 K are shown in Figure 7. The curves for both radicals **3** and **4** are fitted to a monoexponential function with a build-up time *T_B_* of approximately 1 s. Although for radical **3**, this monoexponential fit represents the data points excellently (within error bars), for radical **4**, slight deviations are seen indicating differences in the dynamics of the spin system that are probably attributable to the presence of different conformers in the solution of radical **4**.

#### 2.4.3. Direct and Indirect Polarization in ^13^C MAS

Next, in ^13^C MAS experiments, polarization transfer was investigated, which gives rise to different pathways depending on local dynamics in the surroundings of a radical [12,33,34,35,36,37].

Figure 8 shows the ^13^C MAS spectra obtained for biradicals **3** and **4** with and without microwave irradiation. To take a closer look at the polarization transfer mechanism revealed for the radicals, the presence of the direct polarization pathway from e^−^ to ^13^C—as well as via the indirect polarization pathway from e^−^ to ^1^H and then through cross-relaxation to ^13^C—was analyzed for a build-up time of 300 s. In the µW_OFF_ spectra (Figure 8a,c), only the direct pathway was obtained, which is not surprising because the efficiency of the polarization transfer via cross-relaxation in solids is only moderate [38] and is thus difficult to observe without DNP signal enhancement. In the µW_ON_ spectrum of biradical **3** (Figure 8b), the indirect pathway became prominent, probably under the influence of this radical’s methyl side groups initiating cross-relaxation. On the contrary, the µW_ON_ spectrum of biradical **4** (Figure 8d) does not show the indirect pathway, probably because of the more rigid structure of this radical containing the spiro-substituents, as shown for other rigid radicals [12]. In a comparison of the signal enhancements for the direct pathway obtained for both radicals, a factor of 6 is achieved for radical **3**, whereas a factor of 2 is achieved for radical **4** at a built-up time of 300 s.

### 2.5. The Ring-Opening Reaction

To study the ring-opening reaction under the influence of nucleophiles, we performed the following experiments. To a solution of each biradical at a concentration of 2 × 10^−4^ M in dimethyl sulfoxide, *n*-butylamine was added in an equal volume (a~10,000-fold molar excess). Then, EPR spectra of the resulting reaction mixture were recorded at regular intervals for 3 h at room temperature. We observed a change in the intensity ratio of EPR lines correspondent to biradicals with *J ≈* 0 (lines 1, 3, and 5) and lines correspondent to biradicals with a strong exchange interaction *J* >> 0 (lines 2 and 4). The obtained spectra were simulated assuming two sets of biradicals, namely with *J_1_* = 100 (90 for radical **4**) MHz and *J_2_* = 0 MHz. The contributions to the spectrum from two components with different *J* (weights) were calculated, and the resulting dependences of these weights on reaction time are outlined in Figure 9. For biradical **4**, almost no changes were observed (Figure 9b). The increase in the intensity of the exchange lines (2,4) is most likely due to a change in the conformation distribution upon the addition of the amine. For biradical **3**, a significant decrease in the contribution of exchange lines was noted. For both biradicals, integral intensity barely changed during the reaction, indicating that the number of radical centers remained constant and the radical moieties themselves do not react with *n*-butylamine.

Unlike the examples described earlier [27,39,40], for asymmetrically substituted 1,3-diazetidines, two possible reaction pathways can be proposed for the reaction with amines (Figure 10). Given that the EPR spectra of radical **4** stayed virtually unchanged during the reaction, it can be assumed that in the spiro-substituted biradical, either the ring opening does not occur, or the reaction proceeds along pathway 1 (the resultant distance between two nitroxyl moieties is less in pathway 1 than in pathway 2). Perhaps this phenomenon is due to a steric barrier created by bulky cyclohexyl substituents in pathway 2. For methyl-substituted biradical **3**, both pathways are possible; this property accelerates the ring opening and significantly increases in the contribution of biradicals with *J ≈* 0. The findings from this experiment clearly prove the superiority of spiro-substituted biradicals in a nucleophilic medium.

## 3. Experimental Section

### 3.1. Synthetic and Characterization Procedures

#### 3.1.1. 14-((Ethoxycarbonyloxy)carbonyl)-7-azadispiro[5.1.5.2]pentadeca-14-ene-7-oxyl 6

It was prepared according to a procedure from the literature [24] and used for the synthesis of compounds **8** and **2** (Figure 2). Compounds **1** and **7** were derived from **5** according to this scheme as reported in a previous work [22].

#### 3.1.2. 14-(Azidocarbonyl)-7-azadispiro [5.1.5.2]pentadeca-14-ene-7-oxyl (8)

A solution of anhydride **2** (250 mg, 0.74 mmol) in acetone (10 mL) was cooled to 0 °C, a solution of sodium azide (120 mg, 1.85 mmol) in water (5 mL) was added, and the mixture was stirred for 2 h. After the removal of acetone in vacuum at ~0 °C, the product formed was extracted with EtOAc (5 mL × 4). The combined extracts were dried over MgSO_4_ with vigorous stirring for ~30 min. The solvent was removed under reduced pressure (without heating!) to prepare **6** as an orange oil (187 mg, 87%). IR ν_max_ (neat) 2931, 2858, 2140, 1691, 1618, 1448, 1415, 1359, 1307, 1226, 1189, 1074, 960, 908, 894, 815, 740.

#### 3.1.3. 14-(Isocyanato)-7-azadispiro [5.1.5.2]pentadeca-14-ene-7-oxyl (2)

A solution of carbonyl azide **8** (110 mg, 0.38 mmol) in dry hexane was heated under reflux for 2 h. The reaction mixture was filtered under Ar, and the solvent was removed under reduced pressure to obtain isocyanate **2** (81 mg, 82%): a pale orange solid, mp 44–46 °C (dec.) IR ν_max_ (Diffuse reflectance) 3081, 2941, 2860, 2252, 1681, 1656, 1510, 1442, 1355, 1340, 1257, 1128, 1062, 1008, 906, 860, 840, 775, 644, 595. HRMS (EI/DFS) m/z [M]^+^ calcd for C_15_H_21_N_2_O_2_: 261.1600, found 261.1601.

#### 3.1.4. N2,1-(ferrocene-1,10-diyl)-N4,14-bis-(7-azadispiro [5.1.58.26]pentadec-14-en-7-yl)-2,4-diimino-1,3-diazetidine (4)

A solution of 1,10-bis(triphenylphosphoranylidenamino)-ferrocene (0.099 g, 0.134 mmol) and radical-isocyanate (0.070 g, 0.27 mmol) in dry dichloromethane (5 mL) was stirred at room temperature for 18 h. The solvent was removed, and the residue was purified by column chromatography (hexane: ethyl acetate, 8:1) to isolate the title product as orange crystals. Yield 0.046 g (48%). ^1^H NMR spectra (CDCl_3_, 400.1 MHz): 4.10–4.60 (m, H_Fe_). *ν*_max_ (KBr): 515 (w), 525 (w), 542 (w), 667 (w), 710 (w), 752 (m), 760 (m), 808 (w), 804 (w), 820 (w), 858 (w), 906 (w), 1019 (w), 1032 (w), 1047 (w), 1055 (w), 1134 (w), 1176 (w), 1198 (w), 1215 (w), 1223 (w), 1238 (w), 1257 (w), 1267 (w), 1280 (w), 1340, 1355 (w), 1379, 1427, 1444, 1502 (w), 1626, 1670, 1774 (w), 1790 (w), 2852, 2928, 3090 (w), 3095 (w). Chemical Formula: C_40_H_50_FeN_6_O_2_ microTOF-Q 103: found [M+H]^+^ 703.343, anal. calc. 703.342.

### 3.2. EPR Analysis

All CW EPR spectra were recorded at X-band frequencies (~9.4 GHz) on a commercial Bruker spectrometer, Elexsys E 540 (Bruker Corporation, Billerica, MA, USA). The temperature measurements were performed by means of an ER 4119 HS resonator in conjunction with an ER 4131VT temperature control system. Liquid samples were loaded into 50 μL glass capillaries, and these were placed in quartz tubes. Electron spin resonance spectra were recorded with the following settings: frequency, 9.25 GHz; microwave power, 2.0 mW; modulation amplitude, 0.02–0.10 mT; time constant, 40.96 ms; and conversion time, 20.8 ms. Simulations of solution electron spin resonance lines were carried out in the EasySpin software (5.2.28) [30], which is available at http://www.easypin.org (accessed on 1 December 2021). Quantitative EPR measurements were carried out via a procedure of comparison with a standard TEMPO solution. The measured concentrations of biradicals turned out to be 6% lower than the theoretical ones. This deviation is included in the measurement error by this method; therefore, it can be concluded that the samples consist exclusively of biradical fragments.

### 3.3. Details of Quantum Chemical Calculations

Geometric parameters of biradicals in the triplet state were optimized at the UB97-D3/def2-TZVP level [39,40,41]. Zero-field splitting parameters (ZFS, D, and E/D) were calculated at the ROBP86/def2-TZVP level [42,43], whereas g-tensors and hyperfine splitting constants at the B1LYP/def2-TZVP level [44].

To locate a series of local minima on the triplet potential-energy surface of biradicals, we performed a relaxed surface scan for rotation around C-N single bonds, i.e., the corresponding dihedral angle was varied, while all other variables were relaxed. The relaxed surface scans were performed at the UB97-D3/def2-SVP level. To calculate exchange coupling constants at these local minima, the Broken Symmetry (BS) approach [45] at the B3LYP/def2-TZVP level [46,47] was used. All these computations were performed with the ORCA 4.2.1 software suite [48].

### 3.4. Solid-State DNP Experiments

Solutions (15 mM) of each radical (**3** or **4**) in TCE were prepared, and approximately 25 µL of each solution was placed in 3.2 mm sapphire rotors. These rotors were sealed with a silicon plug to prevent leakage and were closed with a ZrO_2_ drive cap.

All solid-state DNP experiments were conducted on a Bruker Avance III 400 MHz NMR spectrometer equipped with an Ascend 400 sweep-able DNP magnet and a 3.2 mm triple resonance ^1^H/X/Y low-temperature MAS probe at nominal 115 K. The microwave (µW) irradiation was provided by a 9.7 T Bruker gyrotron system operating at 263 GHz. ^1^H MAS, ^1^H→^13^C CPMAS, and ^13^C MAS spectra with and without µW irradiation were acquired at a spinning rate of 8 kHz. ^1^H NMR spectra were recorded with a background suppression pulse sequence [49], accumulating 16 scans and applying a relaxation delay of 4 s.

Build-up curves for ^1^H were obtained under microwave irradiation (µW_ON_) with a saturation-recovery pulse sequence. A saturation pulse train of twenty 90° pulses with 2.3 µs length and 200 µs spacing was applied followed by a 90° detection pulse. For each data point, 16 accumulations were performed. Respective ^1^H→^13^C CPMAS measurements were performed with 128 scans with recycle delay D1 of 1.3 T_B_. A contact time of 2 ms was utilized in these experiments. During data acquisition, high-power proton decoupling was performed using the spinal 64 sequence [50].

^13^C MAS spectra were recorded with the pulse sequence explained in ref. [33] that can be briefly summarized as follows. To ensure defined initial magnetization on ^13^C, in all experiments, a saturation pulse train was applied to the ^13^C channel. To select the direct polarization pathway during the build-up of the ^13^C magnetization, the magnetization on the proton’s nuclear spin was eliminated by applying rotor-synchronized 180° pulses (spacing of 2τ between the pulses) in the ^1^H channel. Then, a 90° pulse was applied in the ^13^C channel for detection. The indirect polarization pathway via the proton spin pool could not be selected directly but was detected together with the direct pathway in experiments where no 180° pulses were applied in the ^1^H channel. To select the indirect channel, a difference spectrum between the ones obtained without 180° and with 180° pulses had to be calculated. These sequences were used to record spectra with (µW_ON_) and without (µW_OFF_) microwave irradiation for both radicals. During the data acquisition, high-power proton decoupling was performed by means of the spinal 64 sequence [50].

To determine the DNP enhancement factors (*ε)*, the signal peak areas obtained in the ^1^H, the ^1^H→^13^C CPMAS, and ^13^C MAS spectra with μW irradiation were divided by the corresponding signal peak areas obtained without μW irradiation.

### 3.5. Electrochemistry

Cyclic voltammetry measurements were carried out in a methylene chloride solution using a p-8 nano potentiostat (Elins, Russia). A three-electrode electrochemical cell with a platinum working electrode (3 mm), a platinum wire as a counterelectrode, and a silver chloride reference electrode was used. Tetrabutylammonium hexafluorophosphate served as a supporting electrolyte. Ferrocene was used as an internal standard.

## 4. Conclusions

Two ferrocene-containing biradicals were characterized as promising DNP agents. It was demonstrated that the replacement of methyl groups by spirocycles near the N-O group leads to an increase in the contribution of conformers having *J* ≈ 0. In this case, the DNP gain decreases in inverse proportion to the contribution of biradicals having the negligible exchange interaction.

It was also found that when attacked by nucleophiles such as amines, diazetidine undergoes a ring-opening reaction, resulting in a decrease in *J*. The presence of bulky spirocyclohexane substituents almost completely inhibits the ring-opening reaction. Therefore, in the design of biradicals as PAs, it is necessary to take into account the dual effect of spiro-substituents: a significant contribution of the component having DNP-hindering *J* ≈ 0 but increased resistance to nucleophiles, which can lead to an increase in the contributions of conformers with *J* ≈ 0. 

## Data Availability

Not applicable.

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
