# Peer review of "Effects of Spiro-Cyclohexane Substitution of Nitroxyl Biradicals on Dynamic Nuclear Polarization"

_molecules, 2022, doi:10.3390/molecules27103252_

Round 1
Reviewer 1 Report
The paper “Effects of spirocyclohexane substituents in a nitroxide biradical 2 containing a ferrocene-1,10-diyl-substituted 1,3-diazetidine-2,4- 3 diimine coupler on the structure, exchange interaction, polari- 4 zation ability, and reactivity” is written by the international team which is well-known in the field of bi-(di-)radical synthesis for the EPR/NMR/DNP applications. For some DNP and EPR applications, especially in high magnetic fields, it is crucial to have a possibility to tune the exchange (J) and dipolar (D) couplings between the radical parts (moieties) keeping other properties (stability, size, toxicity, long electron relaxation times, etc.). The authors claim that they succeed to synthesize practically “ideal” nitroxide-based polarizing agent for the cross polarization or polarization transfer mechanisms by using ferrocene linkage for the spirocyclohexane-substituted nitroxides.
The paper is well organized and well- written. I recommend it for publication after some minor
- Figure 1. In which units (nm, A?) the distance (or it is not the distance?) between the nitroxyls (green numbers) is given?
- Whether the EPR in frozen solution is done to get more pameters? For example, D.
- What about the stability of the biradical? How stable are they? What are the conditions of their storage?
- As concerning possible applications for the in-vivo studies in the animal/human tomography. What is the influence of iron on toxicity and other biological properties? Can the iron be used as additional element for therapy or inspection?
- Because a series of various experiments at different temperatures are done, it is advisable to clarify the temperature at which the experiments were done in the figure captions. For example, at which temperature 1H build-up curves were measured?
- Line 277 – what is built up time (?)
- Why in Table 3 the calculated hyperfine splittings are given in mT and G? Better in mT and MHz, for example.
- Hyperfine splittings are far away from the experimental observed. Can the authors give more comment – such mismatch is only due to the known nitroxide problem [ref 30 and Hermosilla, L., et al. "DFT calculations of isotropic hyperfine coupling constants of nitrogen aromatic radicals: the challenge of nitroxide radicals." Journal of Chemical Theory and Computation1 (2011): 169] or iron-linkage gives more additional error.
Author Response
We are very thankful to reviewer and we took into account all his comments.
The paper “Effects of spirocyclohexane substituents in a nitroxide biradical 2 containing a ferrocene-1,10-diyl-substituted 1,3-diazetidine-2,4- 3 diimine coupler on the structure, exchange interaction, polari- 4 zation ability, and reactivity” is written by the international team which is well-known in the field of bi-(di-)radical synthesis for the EPR/NMR/DNP applications. For some DNP and EPR applications, especially in high magnetic fields, it is crucial to have a possibility to tune the exchange (J) and dipolar (D) couplings between the radical parts (moieties) keeping other properties (stability, size, toxicity, long electron relaxation times, etc.). The authors claim that they succeed to synthesize practically “ideal” nitroxide-based polarizing agent for the cross polarization or polarization transfer mechanisms by using ferrocene linkage for the spirocyclohexane-substituted nitroxides.
The paper is well organized and well- written. I recommend it for publication after some minor
- Figure 1. In which units (nm, A?) the distance (or it is not the distance?) between the nitroxyls (green numbers) is given?
Distances are given in Angstroms (Å). We added explanation in Figure 1 caption.
- Whether the EPR in frozen solution is done to get more parameters? For example, D.
We measured EPR in frozen solution. Due to contribution of exchange interaction and dipole-dipole interaction as well as distribution of biradcals with different configurations and different distances between radical centers, it was impossible to extract parameters of exchange interaction and D unambiguously.
- What about the stability of the biradical? How stable are they? What are the conditions of their storage?
Biradicals are very stable in solid phase and can be kept at room temperature. Their stability in liquid state depends on solvent and is determined by stability of nitroxides. Biradicals are not stable in reducing media and at high concentration of nucleophiles.
- As concerning possible applications for the in-vivo studies in the animal/human tomography. What is the influence of iron on toxicity and other biological properties? Can the iron be used as additional element for therapy or inspection?
We did not investigate the toxicity of synthesized biradicals. The observed DNP efficiency was not very high, thus it is not make sense to check toxicity. The structure of synthesized biradicals should be improved to increase DNP. It could be mentioned that nitroxides have very low toxicity.
- Because a series of various experiments at different temperatures are done, it is advisable to clarify the temperature at which the experiments were done in the figure captions. For example, at which temperature 1H build-up curves were measured?
The explanation how the built-up time is determined is given in the experimental section. We further introduced the following sentence with explanation of built up time:
DNP build-up curves (the dependence of signal intensity on time of microwave irradiation) for 1H at nominal 115 K are shown in Figure 7.
- Line 277 – what is built up time (?)
We introduced the following sentence with explanation of built up time:
“DNP build-up curves (the dependence of signal intensity on time of microwave irradiation) for 1H at nominal temperature 115 K are shown in Figure 7.”
- Why in Table 3 the calculated hyperfine splitting are given in mT and G? Better in mT and MHz, for example.
We have corrected Table 3 according to the comment of review.
- Hyperfine splittings are far away from the experimental observed. Can the authors give more comment – such mismatch is only due to the known nitroxide problem [ref 30 and Hermosilla, L., et al. "DFT calculations of isotropic hyperfine coupling constants of nitrogen aromatic radicals: the challenge of nitroxide radicals." Journal of Chemical Theory and Computation1 (2011): 169] or iron-linkage gives more additional error.
Yes, this is a problem of nitroxides. We changed the previous sentence
“this problem is typical for DFT calculations” to “this problem is typical for DFT calculations of HFI constants in nitroxides30,31 and added new reference recommended by reviewer
- Hermosilla, J. M. Garcıa de la Vega, C. Sieiro, and P. Calle, DFT Calculations of Isotropic Hyperfine Coupling Constants of Nitrogen Aromatic Radicals: The Challenge of Nitroxide Radicals, J. Chem. Theory Comput. 2011, 7, 169–179.
Reviewer 2 Report
In this manuscript, Bagryanskaya and coworkers report on the effects of spiro-substitutions on the ferrocene-1,10-diyl-substituted 1,3-diazetidine-2,4-diimine nitroxyl biradicals, including the exchange interaction, polarization ability, and reactivity. The development of polarizing agents (PAs) holds great promise for improving the sensitivity of solid-state NMR spectroscopy, which plays a curial role in the study of biological systems. Thus, this work will surely be of interest to a wide audience. Although nitroxide biradicals have been extensively studied as PAs, and the nitroxide biradicals containing a ferrocene-1,10-diyl-substituted 1,3-diazetidine-2,4-diimine coupler via a [2+2]-cycloaddition have also been reported by the same group, the principle of spiro-cyclohexane substitution on the same nitroxyl biradical backbone remains unknown. Additionally, the influence of spiro-cyclohexane substitution on nitroxyl biradicals is fully studied by the systemic comparison of methyl and spiro-cyclohexane substituted compounds, including single-crystal XRD analysis, cyclic voltammetry, EPR analysis, quantum chemical calculations, and dynamic nuclear polarization experiments. Overall, the manuscript is highly accessible and logically constructed, and the main conclusions are sound. This paper is suitable for Molecules after some revisions.
- The title of this manuscript is too long and difficult to catch the key findings of this work. The title of this manuscript could be shorted and condensed. A possible title for this contribution is given as follows. “Effects of Spiro-Cyclohexane Substitution of Nitroxyl Biradicals on Dynamic Nuclear Polarization”.
- In the Abstract section, lines 21-22, the authors state that “This paper describes a study on biradicals with a ferrocene-1,1’-diyl-substituted 1,3-diazetidine-2,4- diimine coupler as DNP agents that have never been used before”. Since the nitroxyl biradical containing a ferrocene-1,1’-diyl-substituted 1,3-diazetidine-2,4- diimine coupler (control compound 3) has been reported (ref. 21, Tetrahedron Lett., 2017, 58, 478–481), the description in the abstract needs to be revised and corrected.
- Besides control compound 3, other widely used polarizing agents containing nitroxyl biradicals could also be involved and briefly discussed in order to highlight the spiro-cyclohexane substitution approach for improving the DNP characteristics.
- The CV curve of nitroxide biradicals 3 is strongly suggested to be added in Figure 2 for comparison. In the meantime, the related discussion should be added to the main text in the proper position.
- In the captions of Scheme 1, Figures 1-4, the compound number for 3 and 4 should be bold.
- The caption of Figure 5 should be checked, E is missing.
- The expression of the chemical structure of R= spirocyclohexane is ed in Schemes 1, 2, and Figure 10 confusing.
- The correlation coefficients (R2) of non-linear curve-fitting could be added and inserted in Figure 7.
- The full names should be given at their first appearance of the Abbreviations, such as EPR (line 23), DFT (line 24), etc.
- In the Supplementary Information, page 2, 3rd paragraph, the CCDC numbers of single-crystal 3 and 4 should be added.
Author Response
We are very thankful to reviewer and we took into account all his (her) comments.
In this manuscript, Bagryanskaya and coworkers report on the effects of spiro-substitutions on the ferrocene-1,10-diyl-substituted 1,3-diazetidine-2,4-diimine nitroxyl biradicals, including the exchange interaction, polarization ability, and reactivity. The development of polarizing agents (PAs) holds great promise for improving the sensitivity of solid-state NMR spectroscopy, which plays a curial role in the study of biological systems. Thus, this work will surely be of interest to a wide audience. Although nitroxide biradicals have been extensively studied as PAs, and the nitroxide biradicals containing a ferrocene-1,10-diyl-substituted 1,3-diazetidine-2,4-diimine coupler via a [2+2]-cycloaddition have also been reported by the same group, the principle of spiro-cyclohexane substitution on the same nitroxyl biradical backbone remains unknown. Additionally, the influence of spiro-cyclohexane substitution on nitroxyl biradicals is fully studied by the systemic comparison of methyl and spiro-cyclohexane substituted compounds, including single-crystal XRD analysis, cyclic voltammetry, EPR analysis, quantum chemical calculations, and dynamic nuclear polarization experiments. Overall, the manuscript is highly accessible and logically constructed, and the main conclusions are sound. This paper is suitable for Molecules after some revisions.
The title of this manuscript is too long and difficult to catch the key findings of this work. The title of this manuscript could be shorted and condensed. A possible title for this contribution is given as follows. “Effects of Spiro-Cyclohexane Substitution of Nitroxyl Biradicals on Dynamic Nuclear Polarization”.
We took reviewer comment into account and change the title of paper.
In the Abstract section, lines 21-22, the authors state that “This paper describes a study on biradicals with a ferrocene-1,1’-diyl-substituted 1,3-diazetidine-2,4- diimine coupler as DNP agents that have never been used before”. Since the nitroxyl biradical containing a ferrocene-1,1’-diyl-substituted 1,3-diazetidine-2,4- diimine coupler (control compound 3) has been reported (ref. 21, Tetrahedron Lett., 2017, 58, 478–481), the description in the abstract needs to be revised and corrected.
We corrected the sentence as
“This paper describes a study on biradicals with a ferrocene-1,1’-diyl-substituted 1,3-diazetidine-2,4- diimine coupler that have never been used before as DNP agents”.
Besides control compound 3, other widely used polarizing agents containing nitroxyl biradicals could also be involved and briefly discussed in order to highlight the spiro-cyclohexane substitution approach for improving the DNP characteristics.
We have added references and discussion regarding the use of the spiro-cyclohexane substitution approach for improving the DNP characteristics. «By modifying the radical centers with bulky spirosubstituents the DNP performance could be further increased. Biradicals of the TEKPOL and AMUPol series demonstated the best enhancements, approaching the theoretical limit of DNP enhancement [17,21,22].»
The CV curve of nitroxide biradicals 3 is strongly suggested to be added in Figure 2 for comparison. In the meantime, the related discussion should be added to the main text in the proper position.
We have added the CV curve of biradical 3 and relevant discussion to the main text.
In the captions of Scheme 1, Figures 1-4, the compound number for 3 and 4 should be bold.
It is corrected.
The caption of Figure 5 should be checked, E is missing.
The expression of the chemical structure of R= spirocyclohexane is ed in Schemes 1, 2, and Figure 10 confusing.
It is corrected.
The correlation coefficients (R2) of non-linear curve-fitting could be added and inserted in Figure 7.
The full names should be given at their first appearance of the Abbreviations, such as EPR (line 23), DFT (line 24), etc.
It is corrected.
In the Supplementary Information, page 2, 3rd paragraph, the CCDC numbers of single-crystal 3 and 4 should be added.
It is corrected.